# AI4Boundaries: an open AI-ready dataset to map field boundaries with Sentinel-2 and aerial photography

Raphaël d'Andrimont[1], Martin Claverie[1], Pieter Kempeneers[1], Davide Muraro[1], Momchil Yordanov[1], Devis Peressutti[2], Matej Batič[2], and François Waldner[1]

[1]European Commission Joint Research Centre (JRC), Ispra, Italy
[2]Sinergise, Ljubljana, Slovenia
**Correspondence:** Raphaël d'Andrimont (raphael.dandrimont@ec.europa.eu)

**Abstract.** Field boundaries are at the core of many agricultural applications and are a key enabler for the operational monitoring of agricultural production to support food security. Recent scientific progress in deep learning methods has highlighted the capacity to extract field boundaries from satellite and aerial images with a clear improvement from object-based image analysis (e.g., multiresolution segmentation) or conventional filters (e.g., Sobel filters). However, these methods need labels to be trained on. So far, no standard data set exists to easily and robustly benchmark models and progress the state of the art. The absence of such benchmark data further impedes proper comparison against existing methods. Besides, there is no consensus on which evaluation metrics should be reported (both at the pixel and field levels). As a result, it is currently impossible to compare and benchmark new and existing methods. To fill these gaps, we introduce AI4Boundaries, a data set of images and labels readily usable to train and compare models on field boundary detection. AI4Boundaries includes two specific data sets: (i) a 10-m Sentinel-2 monthly composites for large-scale analyses in retrospect, (ii) a 1-m orthophoto data set for regional-scale analyses such as the automatic extraction of Geospatial Aid Application (GSAA). All labels have been sourced from GSAA data that have been made openly available (Austria, Catalonia, France, Luxembourg, the Netherlands, Slovenia, and Sweden) for 2019 representing 14.8 M parcels covering 376 K $km^2$. Data were selected following a stratified random sampling drawn based on two landscape fragmentation metrics, the perimeter/area ratio and the area covered by parcels, thus considering the diversity of the agricultural landscapes. The resulting "AI4Boundaries" dataset consists of 7,831 samples of 256 by 256 pixels for the 10-m Sentinel-2 dataset and of 512 by 512 pixels for the 1-m aerial orthophoto. Both datasets are provided with the corresponding vector ground-truth parcel delineation (2.5 M parcels covering 47,105 $km^2$) and with a raster version already pre-processed and ready to use. Besides providing this open dataset to foster computer vision developments of parcel delineation methods, we discuss the perspectives and limitations of the dataset for various types of applications in the agriculture domain and consider possible further improvements.

## 1 Introduction

Field boundaries are at the core of many agricultural applications such as mapping crop types and yield estimation. With the development of digital farming platforms, extracting and updating field boundaries automatically has gained much traction

to facilitate customer onboarding. Different spatial and temporal data coverage could be needed according to the desired application.

There are three broad methods to map field boundaries: deep learning, object-based image segmentation and conventional (edge-detection) filters Waldner and Diakogiannis (2020). Deep learning methods can extract field boundaries from satellite/aerial images better than object-based image analysis (e.g. multiresolution segmentation) or conventional filters (Sober filters) because they can learn to emphasize relevant image edges while suppressing others. For instance, Waldner and Diakogiannis (2020) and Waldner et al. (2021) have shown that convolutional neural networks can learn complex hierarchical contextual features from the image to accurately detect field boundaries and discard irrelevant boundaries, thereby outperforming conventional edge filters. More recently, a similar approach has been used to unlock large-scale crop field delineation in smallholder farming systems with transfer learning and weak supervision (Wang et al., 2022). Deep learning methods need labels for training and evaluation. No benchmark data set exists to easily do so. The absence of such benchmark data impedes proper comparison with existing methods. Besides, there is no consensus on which evaluation metrics should be reported (both at the pixel and field levels). As a result, it is currently challenging to benchmark new and existing methods.

Deep learning parcel delineation based on the land parcel identification system has been evaluated in several countries such as France (Aung et al., 2020), Netherlands (Masoud et al., 2019) and Spain (Garcia-Pedrero et al., 2019). However, as there is no European harmonised land parcel identification system, there is no dataset to properly benchmark methods over a variety of landscapes and latitudes.

The Geospatial Aid Application (GSAA) refers to the annual crop declarations made by European farmers for Common Agricultural Policy (CAP) area-aid support measures. The electronic GSAA records include a spatial delineation of the parcels. A GSAA element is always a polygon of an agricultural parcel with one crop (or a single crop group with the same payment eligibility). The GSAA is operated at the region or country level in the European Union's (EU) 28 Member States (MS), resulting in about 65 different designs and implementation schemes over the EU. Since these infrastructures are set up in each region, data are not interoperable at the moment, and the legends are not semantically harmonised. Furthermore, most GSAA data are not publicly available, although several countries are increasingly opening the data for public use. In this study, seven regions with publicly available GSAA are selected representing a contrasting gradient across the European Union (i.e., agricultural system depends on physical and human geography resulting in contrasted landscapes). More detailed information about the GSAA is provided in the section 2.3.

Creating reference AI data sets in remote sensing has been shown to accelerate method development and to help push the boundary of the state of the art. For instance, data sets such as BigEarthNet (Sumbul et al., 2019) and EuroSAT (Helber et al., 2019) have been used for generic land cover. For agriculture, most of the previously published datasets over Europe are focusing on France (BreizhCrop and PASTIS; Rußwurm et al., 2019; Tarasiou et al., 2021) or France and Catalonia (Sen4AgriNet; Sykas et al., 2022). In addition to the fact that no harmonized dataset is currently available for multiple European countries, no dataset combining remote sensing and very high-resolution aerial imagery has yet to be published.

To fill these gaps, we release two AI-ready data sets (pairs of images and labels) for field boundary detection to facilitate model development and comparison:

1. A multi-date dataset based on Sentinel-2 monthly composites for large-scale analyses in retrospect.

2. A single-date dataset based on orthophoto for regional-scale analyses such as the automation of GSAA.

All labels are sourced from public parcel data (GSAA) data that have been made openly available.

## 2   Data and study area

### 2.1   Sampling

The rationale behind this study is to propose ready-to-use data set of Earth observation data with corresponding parcel bound-
aries. Public parcel data are first obtained over several countries/regions (i.e., Austria, Catalonia, France, Luxembourg, Nether-
lands, Slovenia, and Sweden) for the year 2019. After drawing a grid of cells of 4 by 4 km in the ETRS89-extended LAEA
Europe projection (EPSG:3035), a stratified random sampling is drawn based on two variables:

1. the average parcel perimeter/area ratio (PAR) computed for each cell is then distributed over 5 percentile bins.

2. the coverage percentage of parcels within the cell distributed in 10 classes.

We designed a random stratified sampling method to extract the image chips from various landscapes. First, the 4x4 km
grid was overlaid over each country/region where parcel data are available. In each grid cell, the field fraction (in percent) and
the perimeter/area ratio were computed as shown for France in Figure 1. These indicators jointly describe the prevalence of
agriculture (i.e. land proportion covered by agriculture) and the landscape fragmentation (i.e. perimeter area ratio) of each grid
cell. Fifty strata were defined by discretising the field fraction in 10 classes (from 0 to 100% by step of 10%) and the perimeter
area ratio in 5 classes defined by its $20^{th}$ percentile in order to obtain a representative sampling. The goal ws to sample
8,5000 sampling units representing an already important dataset to train deep learning models. To this aim, 170 sampling were
selected per stratum (Figure 2). In those strata where the number of samples is larger than the number of available grid cells,
the sampling units in excess were evenly distributed to the other strata. Within each stratum, grid cells were selected so as to
maximise the balance between source regions.

The resulting sampling results in 7,831 samples distributed as described in Table 1 and in Figure 3.

The samples are mainly distributed from North to South (Figure 3).

### 2.2   Earth Observation data

Image chips of a fixed pixel size are required to feed deep learning models. EO data were extracted for the 2-specific dataset
as shown in Figure 4 :

1. Sentinel-2 monthly composites (March to August 2019) 256 by 256 pixels

2. Orthophoto single-temporal imagery resampled at 1-m resolution 512 by 512 pixels

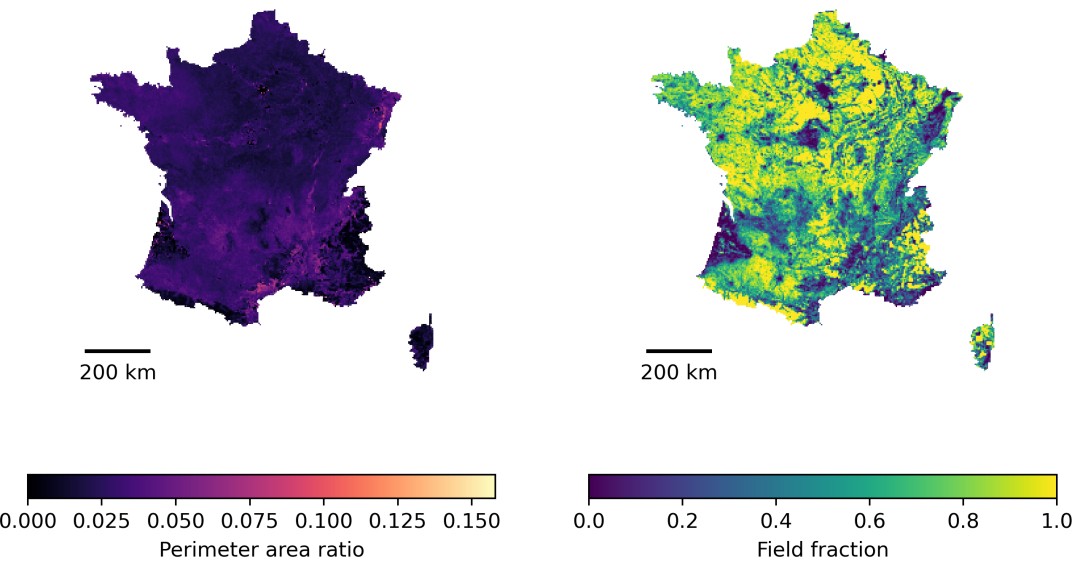

**Figure 1.** The stratification of the sampling is done based on Perimeter area ratio (left figure) and proportion of parcels (right figure) in 4-km × 4-km grid cells. Example across France.

**Table 1.** Distribution of the final stratified sampling for each region.

| Country/Region | Number of sampling units |
|---|---|
| Austria | 2091 |
| Catalonia | 652 |
| France | 2078 |
| Luxembourg | 132 |
| Netherlands | 1157 |
| Slovenia | 301 |
| Sweden | 1420 |
| *Total* | 7831 |

The difference in pixel extent (256 vs 512) of the two datasets is linked to the spatial resolution of Sentinel-2 (10 m) and orthophoto (1 m) respectively corresponding thus to 2560 m by 2560 m and 512 m by 512 m. The extent of the orthphoto extracted has to be extended from 256 to 512 to provide sufficient context.

### 2.2.1 Sentinel-2

This section describes how the monthly cloud-free Sentinel-2 surface reflectance composites for March to August 2019 (thus 6 months of 4 bands: R, G, B; NIR) were produced. The Figure 5 provides an example of the dataset.

The Sentinel-2 Level-2A surface reflectance (SR) were derived from the Sentinel-2 Level-1C top of atmosphere (TOA) reflectance data processed using the Sen2Cor processor (Main-Knorn et al., 2017) from the ESA SNAP toolbox (European

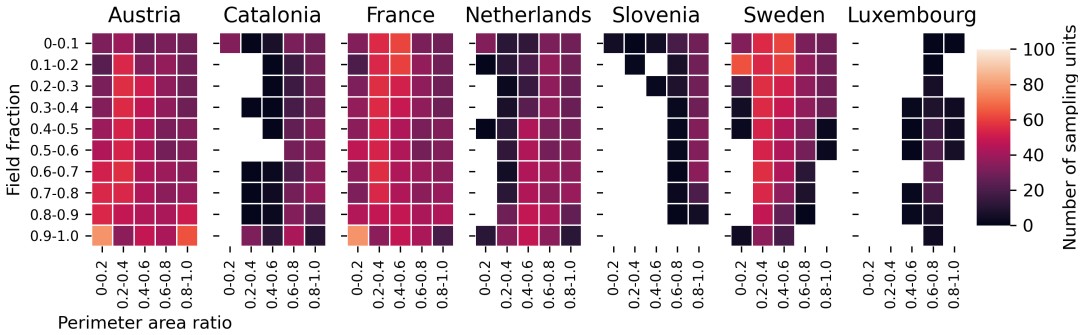

**Figure 2.** Distribution of sampling units among the seven regions.

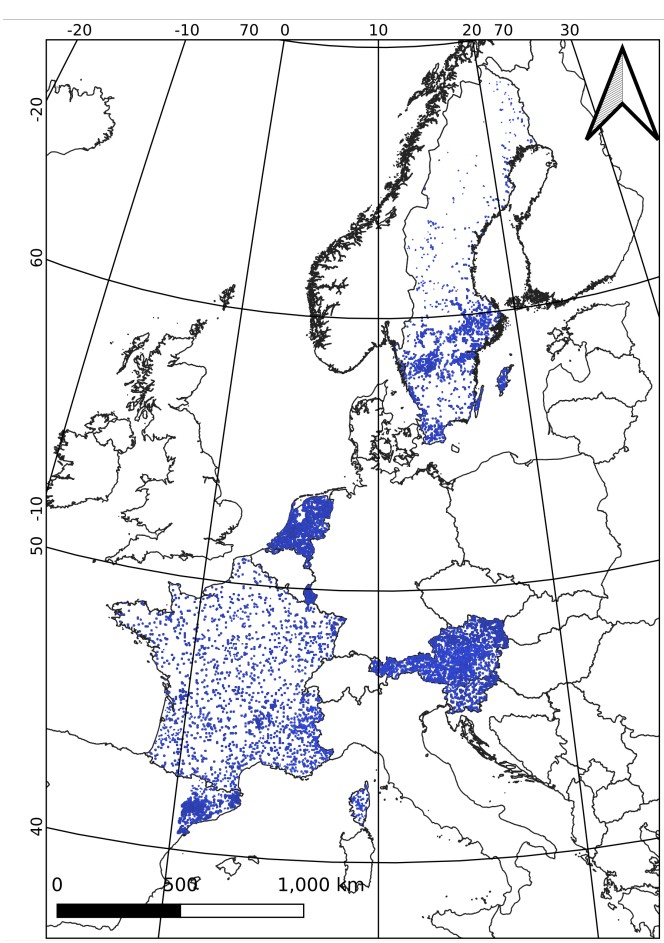

**Figure 3.** Spatial distribution of sampling units among the seven regions.

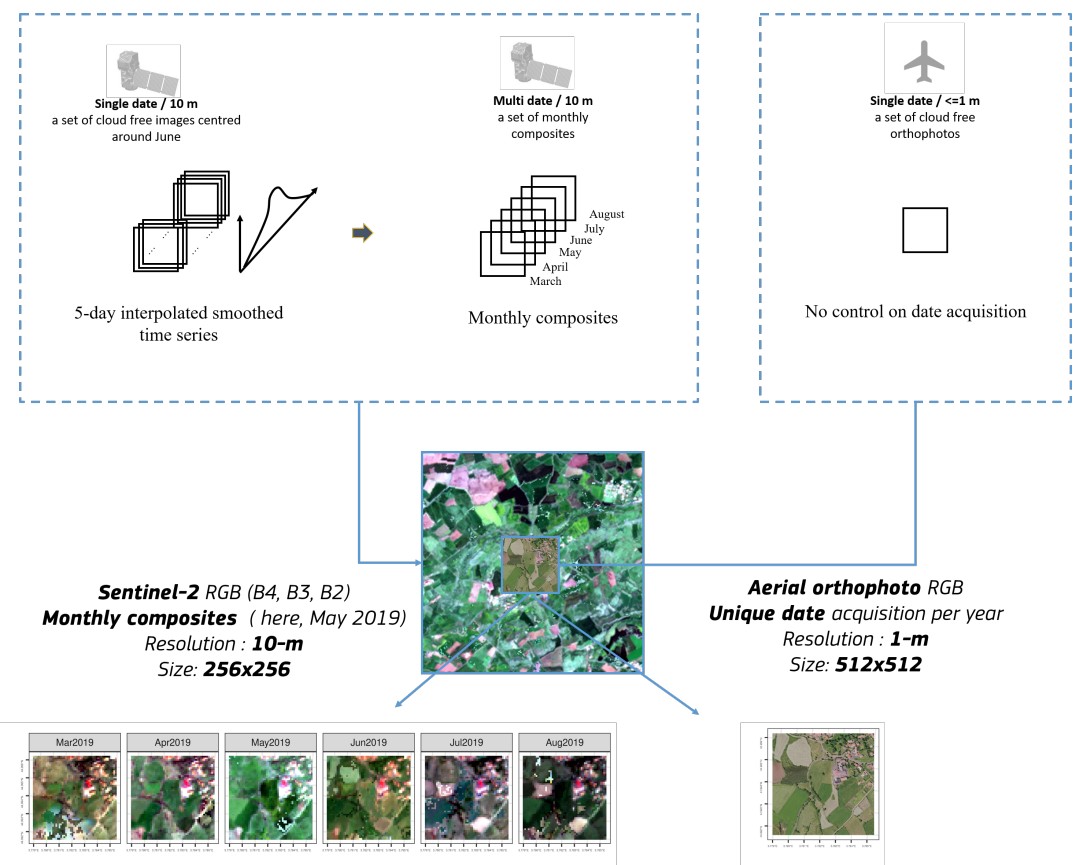

**Figure 4.** Earth Observation generation overview. Sentinel-2 time series are interpolated and smoothed to generate 10-m monthly composites cropped on 256 by 256 pixels. Aerial orthophotos are resampled at 1-m and cropped on 512 by 512 pixels in the center of the 4-km sampled cell. The Sentinel-2 time series cropped to the orthophoto extent is shown in detailed in the Figure A2.

Space Agency). The four spectral bands that are available at a spatial resolution of 10 m were selected (B2, B3, B4, and B8). The Scene Classification Layer (SCL) obtained from Sen2Cor was added as an extra band. Sentinel-2 processing was performed on the BDAP (Soille et al., 2018) using the open-source pyjeo (Kempeneers et al., 2019) Python package.

Data cubes, consisting of merging all 2019 acquisitions for all 4x4 sq. km chips, were created. The Data cubes were extended with the acquisitions of the preceding (December 2018) and successive (January 2020) months. The extra observations served to mitigate the boundary effects at the beginning and end of the time series while applying temporal operations. These months were removed after the filter was applied.

Only pixels identified as dark (SCL=2), vegetated (SCL=4), not-vegetated (SCL=5), water (SCL=6), and unclassified (SCL=7) were considered as "clear". In addition, outliers were detected using the Hampel identifier (Hampel, 1974), based on the pixel values in the red (B4) and near-infrared (B8) bands. The SCL was resampled to 10 m based on the nearest neighbor to obtain a regular gridded data cube. The Hampel filter calculates the median and the standard deviation in a moving window, expressed

as the median absolute deviation (MAD). For the moving window, a width of 40 days was considered. Pixels below two and above three standard deviations from the median were identified as outliers for the NIR and the red bands, respectively. The respective values of two and three standard deviations for the lower and upper bounds were selected ad-hoc based on a visual inspection of the results. The outliers in the red and NIR band domains were used to identify omitted clouds and omitted cloud shadows, respectively. The masked pixels from the SCL and the detected outliers were removed from the time series and replaced by a linearly interpolated time series using "clear" observations; in case of outliers near the beginning and end of the time series, values were extrapolated to the nearest "clear" observation. The resulting time series were then resampled to obtain gap-filled data cubes by taking the mean of the filtered and interpolated values every 5 days.

Despite the SCL masking and the outlier detection, the resulting time series were still noisy. This is due to residual of atmospheric correction and non-accounted bidirectional reflectance distribution function (BRDF) effects. A subsequent smoothing filter was therefore applied, the recursive Savitzky-Golay filter (Chen et al., 2004). The original implementation has been developed for Normalized Difference Vegetation Index (NDVI) time series. It was adapted in this study to smooth surface reflectance values. A total of 15 observations at 5-day temporal resolution were used for the smoothing window size: 7 leftward (past) and 7 rightward (future) observations. The order of the smoothing polynomial was set to 2.

A monthly composite image was then calculated, resulting in 12 observations for the year 2019 for each spectral band (B2, B3, B4, and B8). The composite was calculated as the median value of the smoothed values within each month. The median composite further reduced the remaining noise in the time series by aggregating the observations and reducing the temporal dimension. The median composite was chosen for its robust statistics to outlying observations resulting from atmospheric contamination or phenological variation (Flood, 2013; Brems et al., 2000). In a study on forested areas (?), the median value composites produced the least noisy outputs. More specific on crop classification, the median composite has been successfully applied to Sentinel-1/2 time series in Northern Mongolia (Tuvdendorj et al., 2022).

### 2.2.2 Aerial Orthophoto imagery

Orthophoto extraction was done using public Web Map Tile Service (WMTS) and Web Map Service (WMS) services for the seven regions (see details in Table A1). The orthophotos have a 1-m resolution and an extent of 512 by 512 pixels centred on the centroid of the sampled cells. After downloading a larger extent in the geographic reference system provided by the service, the samples are reprojected to EPSG 3035, cropped to the exact extent and standardised on 3 RGB by removing the NIR band when available to have a consistent dataset. Finally, the histogram extraction of each sample has been used to filter out 233 of the 7,831 samples (corresponding to 232 in Sweden and 1 in France) for which no data is available. Figure 7, along with the vector label data, shows random examples for each country.

The EU context in which aerial photography is collected is specific and has high requirements in terms of spatial accuracy. Indeed, in the EU, the aerial photography campaigns are driven by the need of administration to control the farmers' declarations' validity for aid application.The minimum accuracy requirement is defined in Article 70. of Regulation (EU) 1306/2013 as at least equivalent to that of cartography at a scale of 1:10.000 and, as from 2016, at a scale of 1:5.000. This translates into: (1) a horizontal absolute positional accuracy expressed as RMSE of 1,25m (5.000 x 0,25mm = 1,25m), (2) or the equivalent

**Table 2.** The original dataset contains 14.8 M parcels covering 376 K $km^2$. The stratified sampling resulting in 7,831 4-km samples contains 2.5 M parcels covering 47,105 $km^2$. The mean area refers to parcel area ine hectares, while the total mean area is here in the table the average area for the seven regions.

| Region | Full set | | | Sampling | | |
|---|---|---|---|---|---|---|
| | *Count* | *Area ($km^2$)* | *Mean area ($Ha$)* | *Count* | *Area ($km^2$)* | *Mean area ($Ha$)* |
| *Austria* | 1,631,360 | 31,920.77 | 19.57 | 609,849 | 12,138.88 | 19.9 |
| *Catalonia* | 644,376 | 7,267.31 | 11.28 | 351,403 | 4,589.57 | 13.06 |
| *France* | 9,604,463 | 279,750.40 | 29.13 | 562,568 | 12,613.27 | 22.42 |
| *Luxembourg* | 92,397 | 1,280.16 | 13.85 | 76,657 | 1,044.46 | 13.63 |
| *Netherlands* | 772,565 | 18,686.18 | 24.19 | 399,849 | 8,277.49 | 20.7 |
| *Slovenia* | 820,151 | 4,684.92 | 5.71 | 249,271 | 1,522.60 | 6.11 |
| *Sweden* | 1,282,363 | 32,492.69 | 25.34 | 222,513 | 6,919.22 | 31.1 |
| *Total* | **14,847,675** | **376,082.50** | **25.33** | **2,472,110** | **47,105.48** | **19.05** |

CE95 value, display range and feature type content compatible with a map with a scale 1:5.000 (i.e. topographic maps rather than urban survey maps), (3) using orthoimagery <= 0,5m GSD (see more details in https://marswiki.jrc.ec.europa.eu/wikicap/index.php/Positional_Accuracy ).

## 2.3   Label data

The labels are obtained from vector parcels of the GSAA for each specific region. The GSAA refers to the annual crop dec-
larations made by EU farmers for CAP area-aid support measures. The electronic GSAA records include a spatial delineation of the parcels. A GSAA element is always a polygon of an agricultural parcel with one crop (or a single crop group with the same payment eligibility). The GSAA is operated at the region or country level in the EU-28, resulting in about 65 different designs and implementation schemes over the EU. Since these infrastructures are set up in each region, at the moment data is not interoperable, nor are legends semantically harmonised. Furthermore, most GSAA data is not publicly available, al-
though several countries are increasingly opening the data for public use. In this study, seven regions with publicly available GSAA are selected representing a contrasting gradient across the EU. The Agri-food Data Portal from the Directorate-General for Agriculture and Rural Development references Member State Geoportals providing links where the data could be down-loaded (https://agridata.ec.europa.eu/extensions/iacs/iacs.html). After downloading the dataset, a reprojection to EPSG 3035 was done. From the original set of 14.8 M parcels covering 376 K $km^2$ (Table 2), the 7831 4-km samples contain 2.5 M parcels
covering 47,105 $km^2$ were selected. Finally, for both Sentinel-2 and orthophoto datasets, vector data was rasterised. The label is composed of four bands (example in Figure 6 B, C, D, E): vector label, boundary mask, distance mask and field enumeration. See Figure A3 for detailed overview of the label along with a Sentinel-2 RGB composite.

## 2.4 Train, Validation and Test

We provide the orthophotos Zip archive with their respective masks to be used as a benchmark dataset (see Data availability section to access the data). The split between the samples respects a typical distribution such as training 70%, validation 15% and test 15%. The selection of the sampling is random. This information is stored in the column 'split' of the CSV tables with the URLs of the files. As described previously, 233 samples, almost exclusively in Sweden, have no orthophotos available; thus, the split was done on 7,598 files (7,831 minus 233). The resulting random division provides 5,319 files for training, 1,140 for validation, 1,139 for testing and 233 as NA.

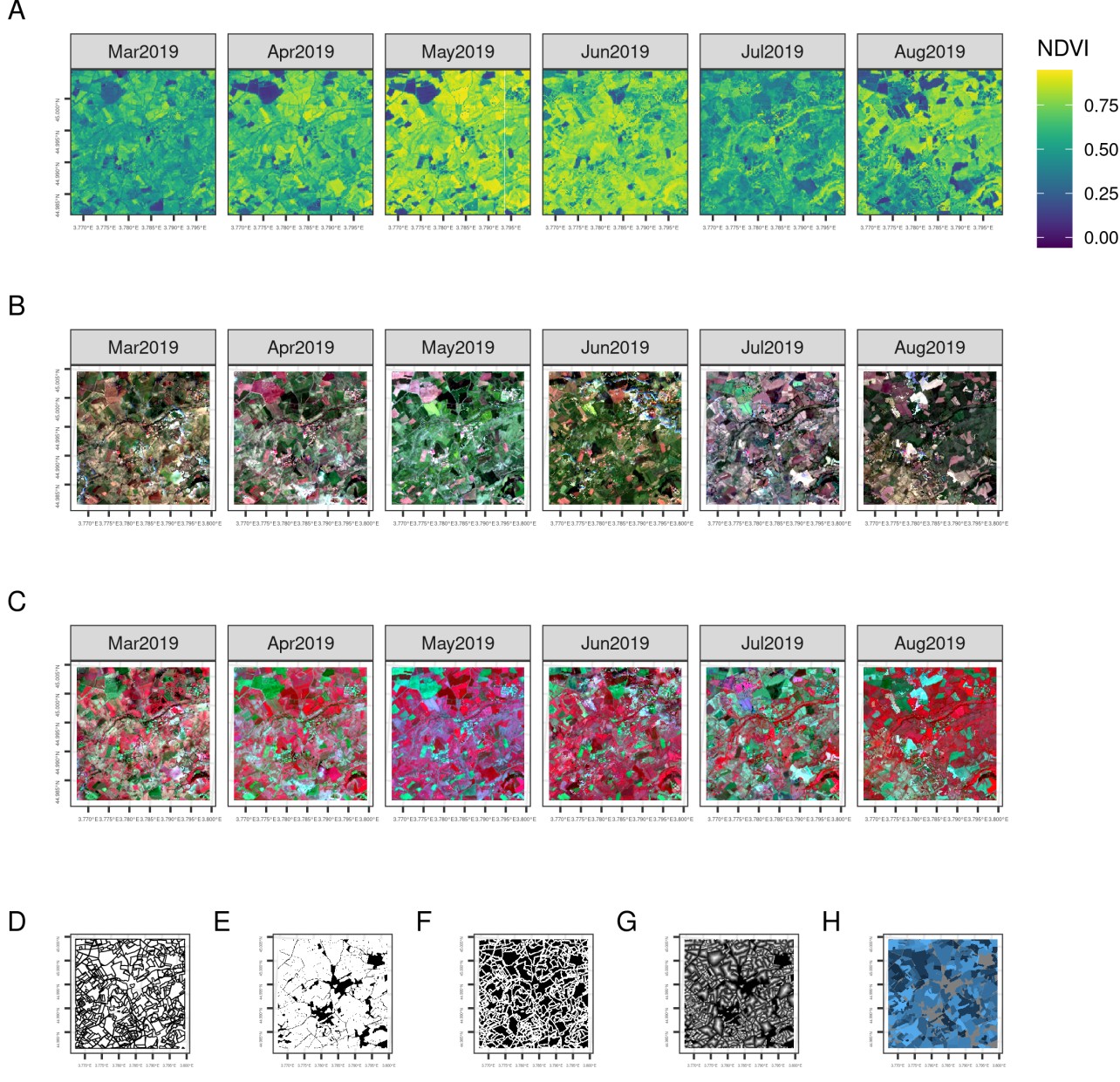

**Figure 5.** Examples of the Sentinel-2 10-m dataset consisting of an extent of 256 pixels of 10-m (thus 2560 m by 2560 m). The samples is located is the South of France (sample id 41781 with the extent coordinates 3827536, 2449682 to 3833405, 2453882 in EPSG 3035) . (A) shows NDVI monthly composite, (B) RGB monthly composites and (C) NIR false colour monthly composites. The vector layer of the label is shown in (D). The label at the same resolution and extent consists of four layers: (E) an extent mask, (F) a boundary mask, (G) a distance mask and (H) a field enumeration. See Figure A3 for more detailed overview.

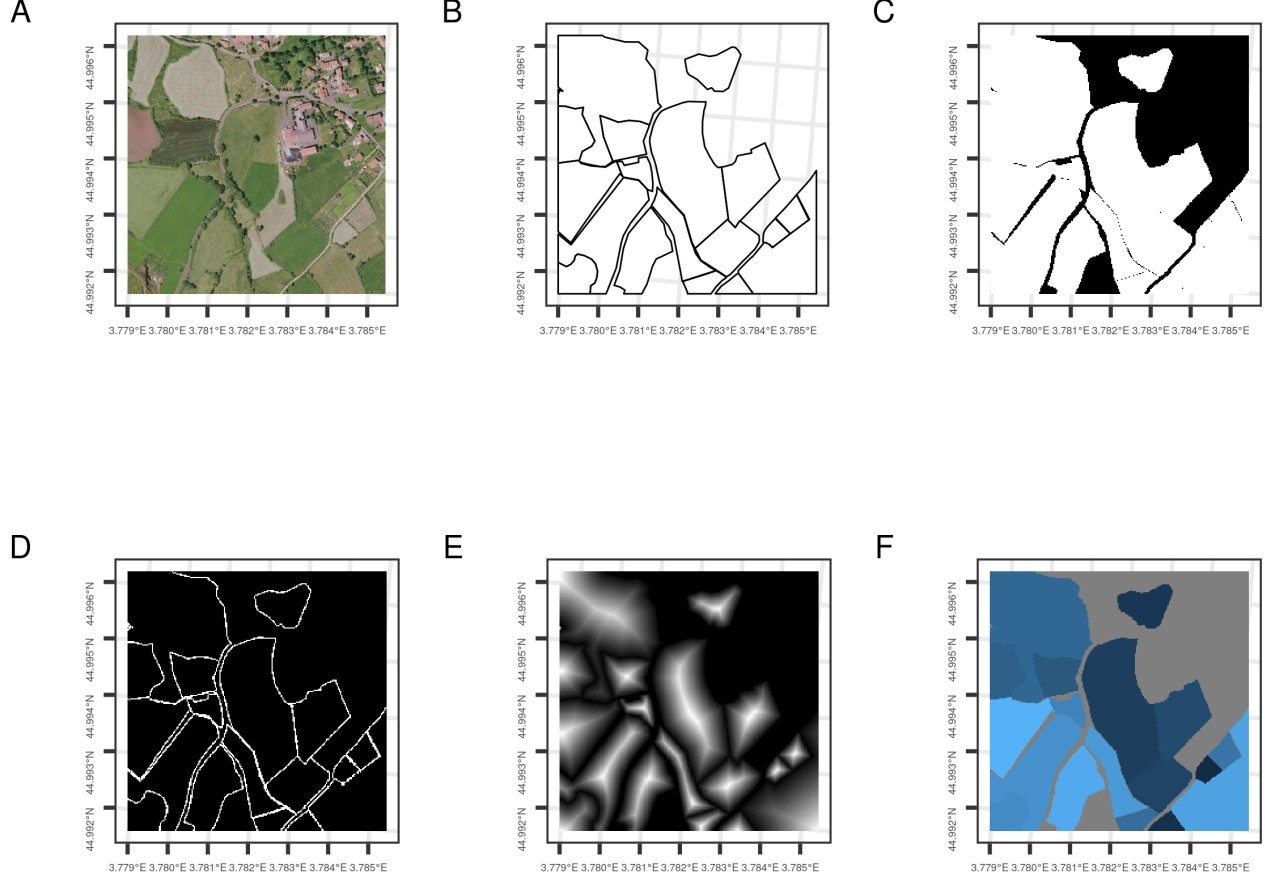

**Figure 6.** Examples of the aerial orthophotos 1-m dataset consisting of an extent of 512 pixels of 1 m (thus 512 m by 512 m). (A) aerial orthophoto RGB. The vector label (B) and the raster label at the same 1-m resolution and extent consist of four layers: (C) an extent mask, (D) a boundary mask, (E) a distance mask and (F) a field enumeration.

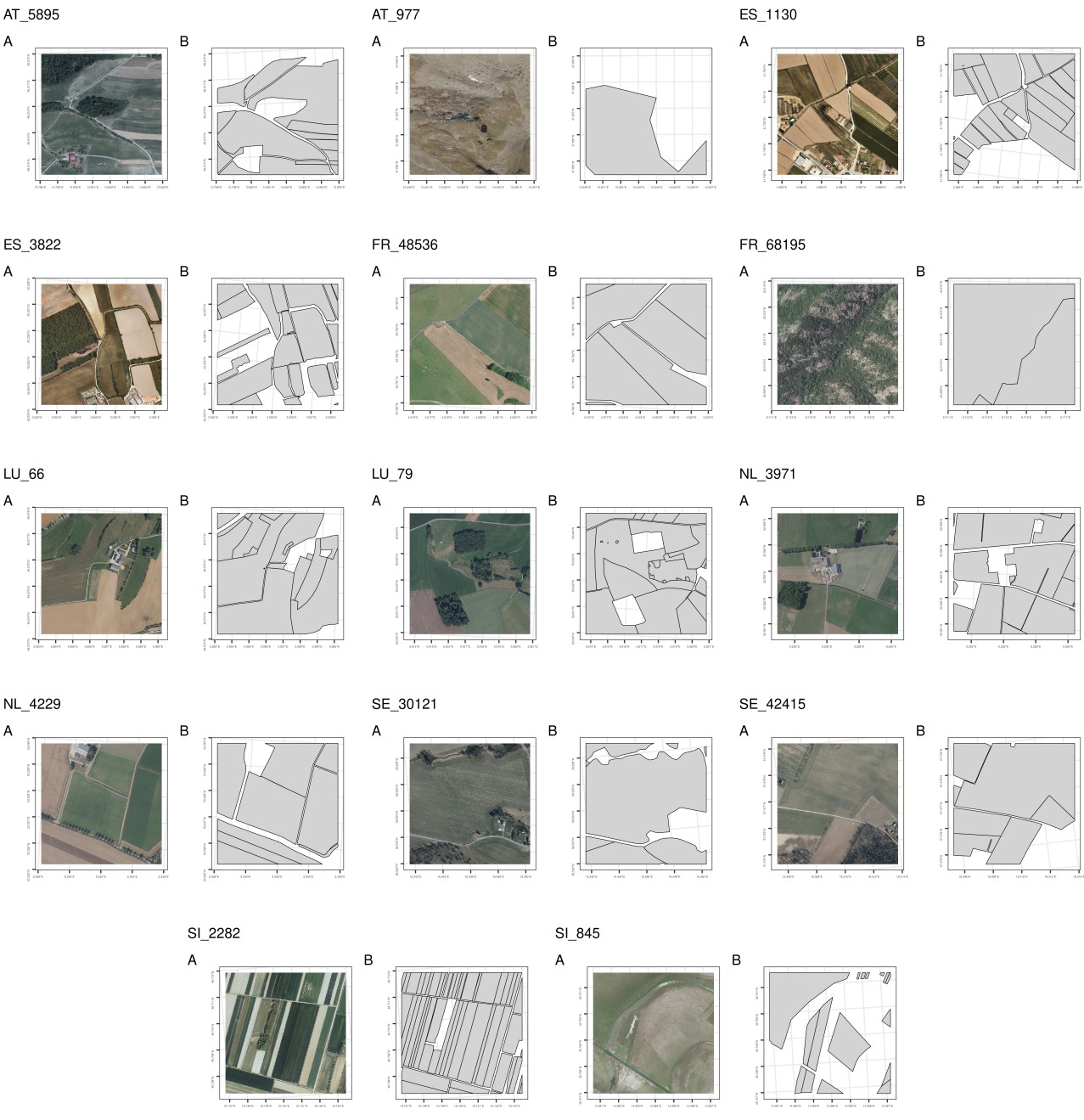

**Figure 7.** Two random aerial orthophotos (A) examples for each country along with the corresponding parcel vector labels (B). File id including NUTS0 (e.g. AT_5895) is above each example.

## 3 Limitations and Perspectives

In this section, we point out some limitations and potential improvements of the approach and the proposed dataset.

The atmospheric corrections and cloud screening remain a challenge for Sentinel-2. We implemented a pragmatic approach to improve the bottom-of-atmosphere reflectance obtained from sen2cor (Main-Knorn et al., 2017). The Hampel outlier detection approach followed by a Savitsky-Golay smoothing allows to produce a 5-day interpolated smoothed data. However, residual cloud, cloud shadow or haze thus jeopardize the development of applications (see Figure A2 where undetected clouds result in artefacts on the time series). From the interpolated data, we obtain a median monthly composite to reduce the data size. This approach also has limitations and we could question the usefulness of interpolating the data if the ultimate goal is to produce a median monthly composite, as it could represent an extra computing burden with a limited added value.

In the regions covered by the dataset, the average size of the parcel is 25.33 Ha ranging from 5.71 Ha in Slovenia to 29.13 Ha in France. Sentinel-2 has also inherent limitations for small parcels monitoring as it was already highlighted (Vajsová et al., 2020). They show that about 10% out of 867 fields less than 0.5 ha in size were not monitorable with Sentinel-2. Of course, parcel delineation is not the same but it gives an idea on the limitations inherent to Sentinel-2 resolution. A good illustrtaion to see the difference of the resolution between the Sentinel-2 and orthophoto datataset is to look at the orthophoto on Figure 6 A and the same location cropped to the same extent on the Sentinel-2 in the Figure A2.

The access to the orthophoto services was done either via WMTS or via WMS. A specific server access has to be used for each country with different projections (most in EPSG:3857 but some in local projections as shown in Table A1). While for most of the country a specific capability layer allows to select the specific year of the service, the specific data of acquisition is most of the time unavailable. Additionally, the data quality is heterogeneous and depends on the specific acquisition.

The labels are obtained from GSAA containing inherent caveats. First of all, the geometry accuracy is referred to as 1/5000, i.e. better than 1 m. Sometimes, parcels do not correspond to the agricultural field. Limitations of the labeled dataset could be the geometries, the timeliness and also the semantics. As agricultural fields might be missing (e.g. due to not being present in original GSAA data), the datasets are really only suitable for the masked approach in training - the models trained on AI4boundaries should only learn about the borders, extent and distance of the included fields.

Several potential improvements have to be considered in the future.

First, in addition to the field boundaries, the crop type could be added to enable semantic segmentation similarly to Sykas et al. (2022). To do it properly over a large scale would require harmonising the legend of the GSAA from the different countries. A recent work (Schneider et al., 2021) has proposed a semantic harmonisation framework for this type of data and could thus serve as basis.

Second, so far the AI4boundaries dataset cover only dataset from EU countries. To support the development of robust algorithm, the dataset should be completed with parcel from other geographical context. This is crucial to have validated and generalisable methods.

Another potential improvement would be to add other data sources such as radar data (e.g. Sentinel-1 coherence) or high-resolution satellite time series such as Planet data.

It is also very important to align such type of dataset with new emerging standards such as the one proposed by Radiant Earth ML HUB (https://mlhub.earth/, Alemohammad (2019)). Their Spatio Temporal Asset Catalog (STAC) helps to make geospatial assets openly searchable and indexable.

A limitation of the proposed AI4boundaries is its availability on an FTP server only, not directly callable as python packaged dataset. Having the dataset accessible similarly to the Crop Harvest (Tseng et al., 2021) or Calisto (https://github.com/Agri-Hub/Callisto-Dataset-Collection) would be more user-friendly.

Finally, the AI4boundaries Sentinel-2 dataset has been used to train a model (based on the work of Waldner and Diakogiannis (2020)) that is available on Euro Data Cube (EDC) as an algorithm for on-demand automatic delineation of agricultural field boundaries over user-defined Area Of Interest (AOI). The algorithm (https://collections.eurodatacube.com/field-delineation/) uses Sentinel-Hub services for accessing Sentinel-2 data, and can be employed to produce baseline results for benchmarking purposes. In future work, the AI4boundaries could be made available directly on EDC along with a tutorial on how to use it to increase the outreach to the community of potential users.

## 4   Conclusions

The AI4boundaries dataset provides a statistical sampling of agricultural parcel boundaries over key regions of Europe along with 10-m Sentinel-2 satellite time series and 1-m aerial orthophoto imagery. This unique dataset allows to benchmark and compare parcel delineation methodologies in a transparent and reproducible way.

*Data availability.* This section describes each data set provided along with this document and downloadable here: https://jeodpp.jrc.ec.europa.eu/ftp/jrc-opendata/DRLL/AI4BOUNDARIES (d'Andrimont et al., 2022):

**./sampling** :

*ai4boundaries_sampling.gpkg* is a geopackage vector file containing the 7831 4-by-4 km polygons of the sampling along with the stratification values as attributes;

*ai4boundaries_ftp_urls_all.csv* is a table that contains the path on the JRC FTP server of each Sentinel 2 tiles, orthophotos, and the respective labels of each. This also contains the split (i.e. train, test, val).

*ai4boundaries_parcels_vector.gpkg* consisting in a vector file (geopackage) with the original parcel boundaries on the 4-km grid cell of the sampling

**./sentinel2** :

**./images** : the folder contains 7 folders - one for each NUTS0 region, amounting to a total of 7831 files named *NUTS0_sampleID_-S2_10m_256.nc*. The files are NetCDF of Sentinel 2 tiles at 10m ground resolution, of 256 by 256 pixels, and containing 5 bands (R, G, B, NIR and NDVI) from March to August 2019.

**./masks** : the folder contains 7 folders - one for each NUTS0 region, amounting to a total of 7831 files named *NUTS0_sampleID_-S2label_10m_256.tif*. The files are Geotiff at 10m ground resolution of 256 by 256 pixels and contain 4 bands.

*ai4boundaries_ftp_urls_sentinel2_split.csv* contains the URLs of the sentinel2 image and corresponding mask files along with the split (i.e. train, test, val).

**./orthophoto** :

**./images** : the folder contains 7 folders - one for each NUTS0 region, amounting to a total of 7598 files named *NUTS0_sampleID_-ortho_1m_512.tif*. the files are Geotiff at 1m ground resolution of 512 by 512 pixels and contain 3 bands (R, G, B) acquired in 2019.

**./masks** : the folder contains 7 folders - one for each NUTS0 region, amounting to a total of 7598 files named *NUTS0_sampleID_-ortholabel_1m_512.tif*. the files are Geotiff at 1m ground resolution of 512 by 512 pixels and containing 4 bands.

*ai4boundaries_ftp_urls_orthophoto_split.csv* contains the URLs of the orthophoto image and corresponding mask files along with the split (i.e. train, test, val).

# Appendix A

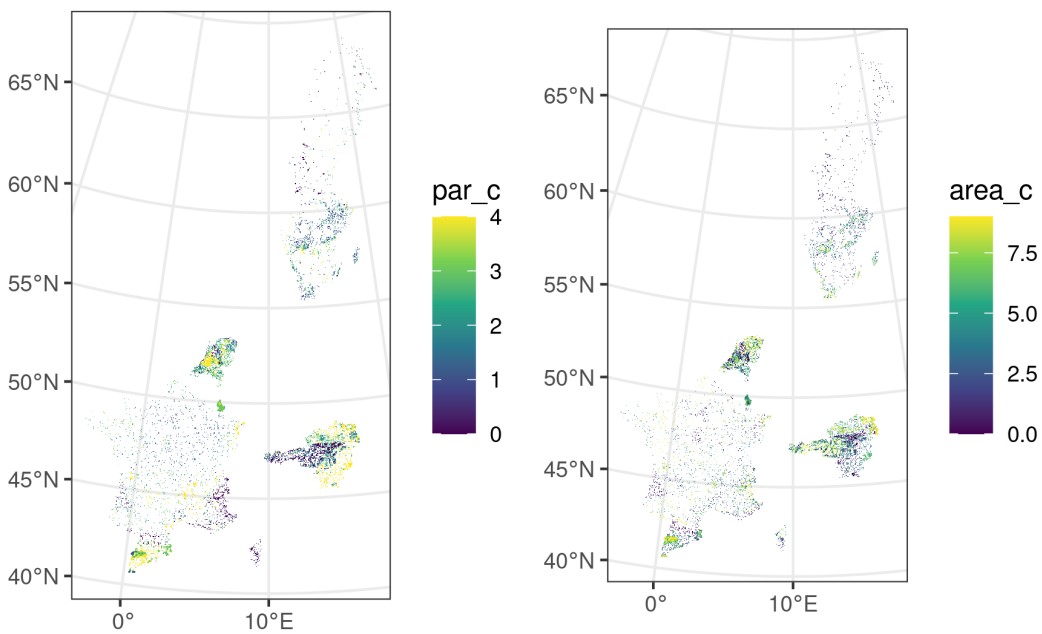

**Figure A1.** Distribution of sampling units among the seven regions with the two variables used for the stratification.

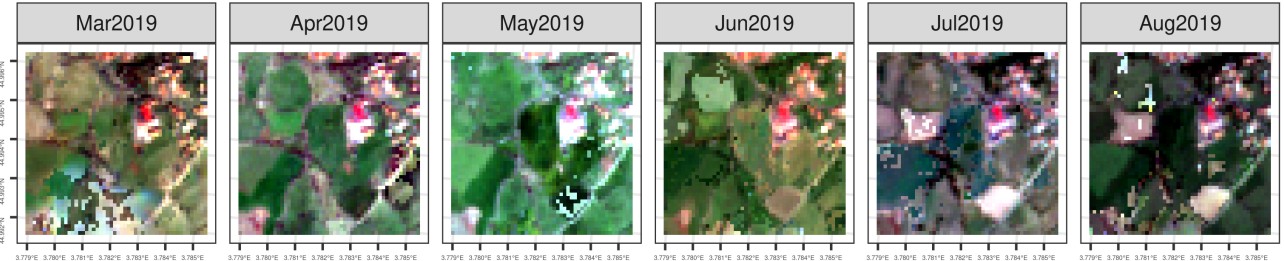

**Figure A2.** Examples of the Sentinel-2 10-m cropped to an extent of the orthophoto data (thus 512 m by 512 m) for ease of comparison.

**Table A1.** Aerial orthophoto WMTS and WMS services and projections.

| country | epsg | capablity layer | type |
|---------|------|-----------------|------|
| Netherlands | 28992 | "https://service.pdok.nl/hwh/luchtfotorgb/wmts/v1_0?REQUEST=GetCapabilities,layer=2019_ortho25" | wmts |
| Luxembourg | 3857 | "http://wmts1.geoportail.lu/opendata/wmts/1.0.0/WMTSCapabilities.xml" | wmts |
| Austria | 3857 | "https://maps.wien.gv.at/basemap/1.0.0/WMTSCapabilities.xml | wmts |
| Catalonia | 3857 | "https://geoserveis.icgc.cat/icc_mapesmultibase/noutm/wmts/topo/1.0.0/WMTSCapabilities.xml" | wmts |
| Slovenia | 3794 | "https://prostor4.gov.si/ows2-gwc-pub/service/wmts?request=GetCapabilities" | wmts |
| France | 3857 | "https://wxs.ign.fr/8ir1y6t0lrcpvtp6up6vc3h7/geoportail/wmts?SERVICE=WMTS&REQUEST=GetCapabilities, layer=ORTHOIMAGERY.ORTHOPHOTOS" | wmts |
| Sweden | 3857 | "https://minkarta.lantmateriet.se/map/ortofoto/" | wms |

*Author contributions.* R.D. and F.W. conceptualized the study and designed the methodology. R. D., M. C., P. K., M. Y., D. P. and F.W. processed the data. R. D., M. C., P. K., D. M., M.Y., D. P. and F.W. analyzed the data and wrote the paper.

*Competing interests.* The authors declare that they have no known competing financial interests or personal relationships that could have appeared to influence the work reported in this paper

*Acknowledgements.* The authors thank Ferdinando Urbano, Guido Lemoine, Marijn van der Velde and Wim Devos for their support.

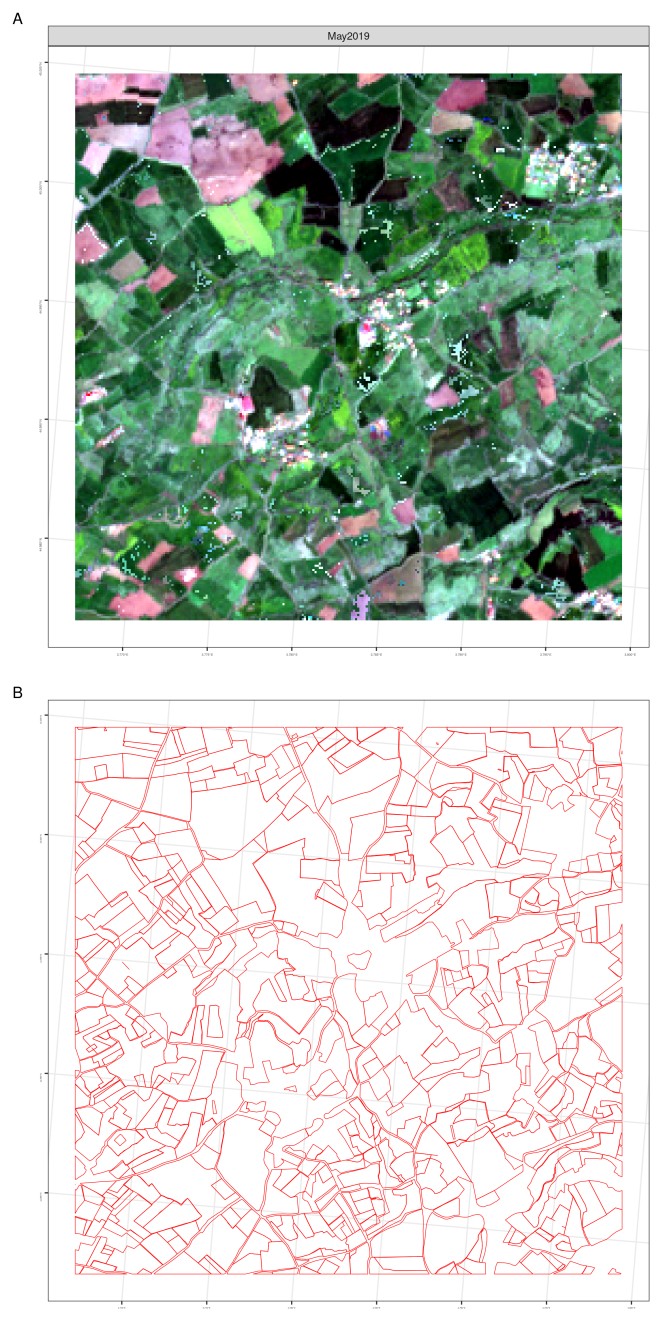

**Figure A3.** Examples of the Sentinel-2 10-m dataset consisting of an extent of 256 pixels of 10-m (thus 2560 m by 2560 m). The samples is located is the South of France (sample id 41781 with the extent coordinates 3827536, 2449682 to 3833405, 2453882 in EPSG 3035) . (A) shows RGB May composites and (B) vector layer of the label.

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
