# Peer review of "AI4Boundaries: an open AI-ready dataset to map field boundaries with Sentinel-2 and aerial photography"

_Earth System Science Data, 2022_

## Author Response (AR1)

**AI4Boundaries: an open AI-ready dataset to map field boundaries with Sentinel-2 and aerial photography**

This document is a detailed reply to reviewers' comments for the AI4Boundaries manuscript.

**Referee #1**

The present study proposed ready-to-use data set of Earth observation data with corresponding parcel boundaries, which can be used to develop a machine learning algorithm for field boundaries delineation and compare methodologies over several countries. The manuscript is well-written and organized. All sufficient information about the developed dataset is presented, so I recommend publishing the present manuscript in its current form.

*We would like to thank the reviewers for their feedback.*

**Referee #2**

The preprint "AI4Boundaries: an open AI-ready dataset to map field boundaries with Sentinel-2 and aerial photography" introduced two data sets for extracting field boundaries (10-m Sentinel-2 monthly composites and 1-m aerial orthophoto), with the labels extracted from the GSAA data. In general, the datasets proposed by the authors could be promising in the algorithm development for parcel boundary detection.

However, I think this preprint needs more attention to get ready for publication. I have several specific comments.

*We would like to thank the referee for the feedback. The manuscript was modified following the remarks and suggestions. Most suggestions have been taken on board. In addition to our responses to each reviewer comment below, a PDF document highlighting the changes between the two versions is submitted.*

1. The GSAA data quality: There is no information about this data, such as how parcel boundaries are derived, their accuracy, etc. The authors may add more information in the introduction section. Further, Figure 6 also shows some missing fields in vector label (B). Although the authors mentioned some limitations of label data, the ability and accuracy should have been addressed in

the beginning to show the potential of this data. Note that this is the main data used for training and testing all ML/DL models to map field boundaries; thus, it has to be highly accurate and confident.

*We thank the reviewer for this valid comment. It is true that highly accurate data is needed to train and test ML/DL model. However, there are inherent limitations in using the GSAA data for this, as, as highlighted by the reviewer some parcels may be missing because not covered by farmers' aid applications. While we are discussing this in the manuscript we are not providing a quantitative assessment of the data quality in the manuscript. Nevertheless, we have added further description related to the GSAA spatial accuracy which should be better than 0.5 m resolution as it is imposed by the EU regulation since 2016. In particular, EU member states have a legal obligation of quality defined by the Article 70, https://eur-lex.europa.eu/legal-content/EN/TXT/HTML/?uri=CELEX:32013R1306 &from=en , see "Identification system for agricultural parcels" stating :*

*"The identification system for agricultural parcels shall be established on the basis of maps, land registry documents or other cartographic references. Use shall be made of computerised geographical information system techniques, including aerial or spatial orthoimagery, with a homogenous standard that guarantees a level of accuracy that is at least equivalent to that of cartography at a scale of 1:10 000 and, as from 2016, at a scale of 1:5 000, while taking into account the outline and condition of the parcel. This shall be fixed in accordance with existing Union standards."*

*Therefore, as of 2016, the spatial accuracy should be better than 1:5 000. We hope that further use of this dataset could also identify further advantages or limitations of using such datasets for ML/DL model development.*

2. The ability of the 10 m Sentinel 2 monthly composite dataset for mapping field boundaries in European countries: A table of field-size statistics is needed. In some areas, field size may be small for large-scale analysis using 10 m data. Do the authors think 10 m data is able to capture small-size fields like in Figure 7? I think it is a critical concern here.

*We thank the reviewer for the comment which is a valid point. We have computed and added the average parcel size for each region in Table 2. Obviously, small parcels could not be delineated with Sentinel-2 dataset. The goal of the AI4boundaries dataset is to propose a dataset for benchmarking and not to do the benchmarking. We have also added a reference in the manuscript on a study "Assessing Spatial Limits of Sentinel-2 Data on Arable Crops in the Context of Checks by Monitoring ". This study highlights the limitations of Sentinel-2 in the European context and shows that 10% of parcels smaller than 0.5 Ha could not be monitored with Sentinel-2. In the regions covered by the dataset, the average size of the parcel is 25.33 Ha ranging from 5.71 Ha in Slovenia to 29.13 Ha in France. Therefore, the size limitations should be limited but of course remains a limitation for very small parcels.*

3. The median monthly composite approach: This seems to lack agreement. What are the current data preparation methods used for field boundary mapping? Do the authors think we may add more noises to the data when we do linear interpolation and smoothing before calculating the median composite? Although the authors provide some insights in the limitations and perspective section, I would suggest adding references that used this composition method for field boundary detection and other applications (e.g., crop mapping) to show the effectiveness of the method.

*The preprocessing steps are specially designed to reduce noises in the time series which are not related to changes of surface themselves. The first operation is the Hampel filter which contributes to removing time series outliers, mainly related to omitted clouds (Hampel filter on Red band) and cloud-shadows (Hampel filter on NIR band) from the SCL layer. These gaps are then filled using linear interpolation, producing an equidistant time series (5 days). The second one is a smoothing process, which aims to remove time series noises related to mis-correction of the atmospheric effects and BRDF effects. The smoothing process involves an adaptive Savitzky-Golay filter. Finally, the median composite is calculated, which further reduces the remaining noise in the time series by aggregating the observations and reducing the temporal dimension. The median composite is chosen for its robust statistics to outlying observations resulting from atmospheric contamination or phenological variation (Brems 2000, Potapov 2011). In a study on forested areas (Potapov 2011), the median value composites produced the least noisy outputs.*

*Four references were added (Flood2013, Brems2000, Potapov2011, Tuvdendorj2022) and the effectiveness of the median composite was augmented:*

*"The median composite further reduced the remaining noise in the time series by aggregating the observations and reducing the temporal dimension. The median composite was chosen for its robust statistics to outlying observations resulting from atmospheric contamination or phenological variation (Flood 2013, Brems 2000). In a study on forested areas (Potapov 2011), the median value composites produced the least noisy outputs. More specific on crop classification, the median composite has been successfully applied to Sentinel-1/2 time series in Northern Mongolia."*

4. It is worth expanding the algorithm development for field boundaries detection in Europe and other countries like the US, their advantages, disadvantages, data, accuracy, etc. This is important to lead the use of Sentinel-2 composite data for large-scale analysis and median composition method. Is there a difference if we use only "clear" for median compositing and then perform interpolation later to fill gaps?

*Expanding spatially the algorithm developement for field boundaries is indeed essential and this AI4boundaries dataset represents a contribution to the community. We have added a paragraph in the discussion about this extension.*

*The second part of the reviewer's remark questions the suitability of the compositing method for Sentinel-2 for larger scale analysis. To address this point, a proper study with a global sampling should be considered and tested. While it would be very valuable, this is outside the scope of the proposed dataset.*

5. For aerial orthophoto and labels: we need a deeper understanding of matching them with satellite Sentinel-2 images instead of only using reprojection. We know that the accuracy of geolocations of aerial orthophoto could be less accurate than well-operated satellites. So, it is also a vital concern to provide accurate labels for training and testing (Sections 2.2.2 and 2.3).

*The EU context in which aerial photography is being collected is specific and has high requirements in terms of spatial accuracy. Aerial photography campaigns are driven by the need of administration to control the farmers' declarations' validity.The minimum accuracy requirement is defined in Article 70. of Regulation (EU) 1306/2013 as at least equivalent to that of cartography at a scale of 1:10.000 and, as from 2016, at a scale of 1:5.000. This translates into:*

- *a horizontal absolute positional accuracy expressed as RMSE of 1,25m (5.000 x 0,25mm = 1,25m), or the equivalent CE95 value,*
- *display range and feature type content compatible with a map with a scale 1:5.000 (i.e. topographic maps rather than urban survey maps),*
- *using orthoimagery <= 0,5m GSD.*

*More information available here: https://marswiki.jrc.ec.europa.eu/wikicap/index.php/Positional_Accuracy*

*We have added a paragraph about it in section 2.2.3 along with the reference.*

6. Besides, the manuscript lacks citations to support its statements. For example, a statement in lines 25-26 needs citations, the GSAA needs a link to the source, etc. Therefore, I encourage the authors to go through the paper and add citations, links to sources, and evidence to support the statements.

*A reference was added in the lines 25-26 and we have carefully reviewed the manuscript to add missing references.*
*For the GSAA sources, the following sentence including reference to where the parcel data could be downloaded has been added in the section 2.3: "The Agri-food Data Portal from the Directorate-General for Agriculture and Rural Development references Member State Geoportals providing links where the data could be downloaded (https://agridata.ec.europa.eu/extensions/iacs/iacs.html)"*

7. Many abbreviations without pre-definition include EO, BDAP, NDVI, WMTS, WMS, EU, ESA SNAP, etc. It may be easy for some users, but it could be hard to understand for others from different backgrounds, such as computer science.

*We thank the reviewer for the comment, we have added the abbreviation definitions missing in the manuscript.*

8. Section 2.4: Is it a random split for all samples? Is it better to randomly split for each country with a similar percentage for training, validation, and testing?

*While the sampling is a stratified random sampling, the subsequent split is random. We have clarified it in the manuscript.*

Line comments:

le with 10,000 units of parcel? Why do we have 170 per stratum?

*We thank the reviewer for spotting this caveat in the manuscript. We initially aimed for 8,500 sampling. For the stratification, we have 10 classes for field fraction and 5 classes for perimeter area ratio. In total, we thus have 50 strata. For each strata we aim to collect about 170 stratum as for some specific stratum no sampling is possible in some countries. If sampling was possible in every country for every 50 stratum, we would have reached 8500 samples.*

Line 82: Explain more about the distribution of samples for each country. It seems like we have denser samples in Austria and Netherlands. (Figure 3 needs a background map of the countries' boundaries).

*For the countries for which we could not sample in a given strata among the 50 strata, we distribute the remaining samples evenly on countries for which the strata is available. This thus resulted in more sampling in some countries for which the strata are available such as the Netherlands and Austria which have sampling available in almost all the 50 strata.*

*Figure 3 was redone and country borders have been added.*

Lines 84-85: Not clear

*This sentence has been rewritten.*

Lines 89-90: This sentence is not clear as well. "The number of pixels for orthophoto data set had to be increased" from … to …?

*Indeed, this was unclear, so we have rephrased the sentence.*

Lines 94-98: We already have ready-to-use surface reflectance (BOA-Level 2A). Why do we use TOA?

*We assume the reviewer refers to the availability of L2A data processed by ESA. As this processing-level is not systematically done by ESA, the data used in for this data set were derived from BDAP processing. The processing is identical from the one done by ESA. However, we recognized the sentence was unclear and we proposed the following modification :*

*"The Sentinel-2 Level-1C top of atmosphere (TOA) reflectance data were obtained and processed using the Sen2Cor processor (Main-Knorn et al., 2017) from the ESA SNAP toolbox to generate the surface reflectance (SR)." replace by "The Sentinel-2 Level-2A surface reflectance (SR) were derive from the Sentinel-2 Level-1C top of atmosphere (TOA) reflectance data processed using the Sen2Cor processor (Main-Knorn et al., 2017) from the ESA SNAP toolbox"*

Lines 40-48: Worth to add more information about the GSAA data as my first main concern.

*We have added a reference to the section 2.3 there where more information is provided about the GSAA used in the manuscript.*

Lines 49-52: Contents look quite similar in lines 32-35

*Agreed, lines 49-52 commented out.*

Line 69: Add more information. Why do we need to use stratified random sampling rather than other sampling methods?

*The stratified random sampling is the best way to have representative sampling maximising the diversity of the landscape. As deep learning models are sensitive to patterns, this will allow to train better models.*

Lines 77-80: It is hard for me to follow. "A sample of 10,000 units…" means only one samp

*The manuscript has been improved.*

Figure 4: Can we separate Sentinel-2 RGB and aerial orthophoto (don't overlap them) and scale them up to clearly see the fields? It is good to compare two data sets to understand the field boundary characteristics.

*Agreed, a new version of the figure is provided adding a zoom in the Sentinel-2 scene. In addition to the new Figure 4, we also provide a detailed figure showing the Sentinel-2 time series cropped on the orthophoto extent to ease the comparison, see in the appendix Figure A2.*

Lines 103-104: The SCL is 20 m spatial resolution. Did the authors resample the SCL band to 10 m and then match it with 10 m bands?

*We have added the following sentence to clarify:*

*The SCL was resampled to 10m based on the nearest neighbour to obtain a regular gridded data cube.*

Figure 5: This figure is important, but subpanels are too small to compare the Sentinel-2 field boundaries with their labels. The coordinates are too small, missing where it is (which country?). Missing D explanation in the caption. Scale sizes should match with each other (D-H do match with A, B, C).

*We thank the reviewer for the comment. We have worked hard to make sure to have an intelligible overview of the dataset in one Figure and, we believe, it is very important to keep it as one figure. The coordinates font are indeed small but readable when zooming in, this is the reason we prefer to keep them.*

*We have added the coordinates of the extent in the caption of the figure. The sample is located in the South of France (sample id 41781 with the extent coordinates 3827536, 2449682 to 3833405, 2453882 in EPSG 3035).*

*We have added the D panel explanation.*

*The size of the panel is (D-H versus A, B, C) indeed does not match because they do not have the same number of panels ( the time series covered 6 months and thus 6 panels while the label covers 5 bands and thus five panels. Nevertheless we value the comment of the reviewer about the difficulty to compare the Sentinel-2 imagery with the label at this scale. To address this comment, we have added a new figure to compare the Sentinel-2 data with the label. See figure A3.*

Figure 6: Caption - "Example of the aerial ortophoto 10-m dataset…" is "Example of the aerial orthophoto 1-m dataset…"? The figure needs a side map showing the chip's locations.

*We thank the reviewer for the remark as indeed, there was a mistake in the caption.*

Line 141: B is the vector label. Is it "(example in Figure 6 C, D, E, F)"?

*We thank the reviewer for the remark and have corrected the sentence.*

Figure 7: Similar comments as Figure 5. I may have misunderstood the paper concept, but I feel it is crucial to compare 10 m Sentinel-2 chips, 1 m orthophotos, and labels to show their correspondence. This is because (i) Sentinel-2 monthly composite is proposed for large-scale field boundaries mapping, and (ii) ML/DL algorithms need labels for training and testing, but currently, I did not see the evidence between Sentinel-2 chips and labels.

*To address this, we have provided two new figures. The Figure A2 is providing Sentinel-2 samples cropped to the orthophoto extent. The Figure A3 provides a detailed overview of the Sentinel-2 with the corresponding labels. We hope that these new elements better support the comparison.*

Line 147: Add where the data are. (Data availability section)

*We have added a reference to the data availability section.*

Table A1: Some links do not work.

*All the links were checked and corrected.*